# Spatially-Explicit Prediction of Capacity Density Advances Geographic Characterization of Wind Power Technical Potential

Dylan Harrison-Atlas * , Galen Maclaurin and Eric Lantz

National Renewable Energy Laboratory, Golden, CO 80401, USA; Galen.Maclaurin@nrel.gov (G.M.);
Eric.Lantz@nrel.gov (E.L.)
* Correspondence: dylan.harrisonatlas@nrel.gov

**Abstract:** Mounting interest in ambitious clean energy goals is exposing critical gaps in our understanding of onshore wind power potential. Conventional approaches to evaluating wind power technical potential at the national scale rely on coarse geographic representations of land area requirements for wind power. These methods overlook sizable spatial variation in real-world capacity densities (i.e., nameplate power capacity per unit area) and assume that potential installation densities are uniform across space. Here, we propose a data-driven approach to overcome persistent challenges in characterizing localized deployment potentials over broad extents. We use machine learning to develop predictive relationships between observed capacity densities and geospatial variables. The model is validated against a comprehensive data set of United States (U.S.) wind facilities and subjected to interrogation techniques to reveal that key explanatory features behind geographic variation of capacity density are related to wind resource as well as urban accessibility and forest cover. We demonstrate application of the model by producing a high-resolution (2 km × 2 km) national map of capacity density for use in technical potential assessments for the United States. Our findings illustrate that this methodology offers meaningful improvements in the characterization of spatial aspects of technical potential, which are increasingly critical to draw reliable and actionable planning and research insights from renewable energy scenarios.

**Keywords:** wind power; capacity density; technical potential; renewable energy; machine learning; geospatial

## 1. Introduction

### 1.1. Problem Overview

Clean technologies are reshaping the energy landscape. Wind technology advancement and learning by doing are driving down costs [1] and fostering growing deployment around the world. In the United States (U.S.), state and federal clean energy policies have further facilitated adoption, resulting in major increases in installed capacity over the past two decades [2]. Continued growth in wind power is expected based on the improving economics [3] as well as increasingly ambitious clean energy targets in many states [4]. As interest in deep decarbonization expands, potentially driving order of magnitude increases in additional wind deployment relative to current levels [5,6], an urgent need to critically evaluate wind energy potential at local to national scales has emerged [7,8].

Investigations into high renewable energy penetration futures have traditionally focused on technical and economic aspects of deployment viability [9,10]. These research thrusts have illuminated the techno-economic conditions that enable broad-scale deployment of wind power. However, sufficient understanding of the central drivers of spatial deployment patterns remains a key limitation as comparably less attention has been paid to the geographic dimension of the energy transition [11,12]. Specifically, expansion of wind energy poses challenges from the perspective of energy planning, which requires a robust accounting of technical potential across time and space [13].

Moreover, perceptions of low energy density of wind power relative to sources that use conventional fossil fuels [14,15] contribute to narratives around expansive wind power footprints and attendant land requirements. The nature of wind energy requires sufficient spacing between turbines and siting around other land uses—such as agriculture, occupied structures, and roads—which, in turn, results in larger footprints for wind energy facilities compared to photovoltaic or conventional generation plants. These siting constraints vary substantially across the country and there remains considerable uncertainty in the land area requirements of wind power when viewed from a national perspective.

Improved understanding of land area requirements is critically needed to assess geographic potential [16] as well as to evaluate potential local and cumulative impacts of wind power deployment and to contrast those against impacts from alternative land uses. During the past decade, significant advancements have been made in modeling wind energy potential. The development of more accurate, higher fidelity representation of wind resource [17] and technology [18] have resulted in better estimations of wind energy generation potential. Moreover, studies increasingly make use of high-resolution data on spatial exclusions [7,19,20], providing greater granularity in the accounting of deployment constraints and barriers.

Although estimates of technical potential have generally improved over time, insights into the geographic variation of this potential remain limited, in part, because of persistent challenges with the representation of spatial characteristics of wind technology deployment [7]. In particular, simplistic assumptions regarding land area requirements for wind power, measured in terms of areal capacity density, are routinely used to determine the amount of nameplate capacity attainable for a given land area. These assumptions stand in contrast to observed sizable variance in reported areal capacity densities (hereafter referred to as capacity density) for onshore utility-scale wind farms [21–23] and contribute to uncertainty in the quantification of technical potential that is sensitive to land area requirements. However, in the absence of granular data or models for inferring how land requirements vary across space, broad geographic generalizations about capacity density have been established as the de facto approach for estimating how much wind power may be obtained over a given area.

These practices limit the understanding of wind energy technical potential with propagation of geographically uniform capacity density assumptions creating further downstream implications for regional energy planning and developing state renewable portfolio standards (RPS). For example, assumed geographically uniform densities at the national scale may either overestimate or underestimate the available capacity, misrepresenting localized opportunities for wind energy that inform the state, regional, and national potential [24]. Estimates of technical potential that are based on a single national average capacity density may be particularly unreliable for applications that require spatially robust information including interregional grid integration studies and power system improvement plans.

With wind energy projected to undergo significant geographic expansion over the next several decades [25], more precise and localized representation of land area requirements will be increasingly needed to draw reliable planning insight from high renewable energy penetration scenarios [26]. Overcoming these challenges requires an alternative modeling approach to produce spatially-explicit predictions of capacity density that reflect localized capacity potential as a function of wind technology, resource, and geospatial variables. In addition to directly informing technical potential, these data-driven insights into wind power capacity density would notably benefit other dimensions of sustainability that relate to the footprint of renewable energy sources. For instance, improved representation of capacity density is needed to better understand the land area constraints of wind versus other clean energy technologies [21]. This information also offers direct insight into competing land occupancy between wind farms and other requirements (e.g., urban growth, conservation, and agricultural) and could illuminate broader ecological and social implications of wind development.

### 1.2. Aim and Contributions

In the present work, we address knowledge gaps in spatial aspects of capacity density by proposing a novel machine learning technique to predict localized capacity densities based on geospatial variables. In contrast to conventional practices that assume capacity density to be uniformly distributed (i.e., a single capacity density) at the national or regional scale, we implement a new approach to explicitly model geographic variation in capacity density at high spatial resolution. Specifically, we trained and validated a machine learning model using geospatial variables and based on observed capacity densities at wind farms located throughout the United States and applied it to create a national map of predicted capacity density. Our primary objective was to map the spatial variability of capacity density as driven by geographic factors across large areas including national and potentially continental scales. Secondary objectives included cultivating deeper understanding of capacity density as it relates to spatial drivers of wind energy deployment patterns and establishing geospatial machine learning methods to advance future investigations of renewable energy technical potential. In addition, we provide an in-depth examination of regional differences in predicted capacity density, offering insights into the driving tvariables.

Our primary contributions are to present a novel methodology for quantifying spatially-varying wind power capacity density at national and continental scales. Our predictive method is unique in that it enables examination of geographic variability in capacity density at high resolution in contrast to existing approaches that assign uniform land requirements across broad geographic extents. To accomplish this, we use machine learning to establish predictive relationships between observed capacity densities at operational wind plants and geospatial drivers. We present an overview of the dominant methodologies that assume spatially uniform capacity densities and differentiate our work by advancing a quantitative method for predicting localized capacity densities that reflect variation in wind power nameplate capacity installation potential. This study exposes the potential range of spatially varying capacity densities across the United States. The observed variability is expected to have a large impact on a wide range of applications including technoeconomic assessments, future capacity expansion modeling, grid integration studies, technology R&D investment, and ecological impact assessments. Moreover, policy decisions made by local, state, and federal actors are best served by a detailed understanding of the quantities and qualities of available wind energy capacity. The enhanced characterization of land use requirements and influential variables (in determining these requirements) detailed here will allow more effective decisions to be made around wind energy deployment and utilization within both the research and policy communities.

## 2. Background

Assessments of technical potential are routinely conducted to evaluate wind energy potential at regional to global scales. These studies provide decision makers with insights into the spatial distribution of potential renewable resources along with an upper-bound estimate of the technically feasible supply that "represents the achievable energy generation of a particular technology given system performance, topographic limitations, environmental, and land-use constraints" [27]. Although technical potential modeling is a data-driven process, it is subject to assumptions about resource, technology, and the environment [28]. Notably, these broad scale estimates of potential differ substantially from individual wind plant siting decisions, which utilize a variety of plant based optimization approaches to balance competing objectives spanning resource, technology, permitting, and costs.

### 2.1. Terminology and Derivation of Capacity Density

A fundamental component of wind energy technical potential is capacity density, which represents the nameplate capacity (i.e., megawatts (MW)) that can potentially be installed over a given area (typically measured in MW/km$^2$). For an individual wind farm, capacity density is computed by summing the installed turbine capacities and dividing by the wind farm area. In practical terms, capacity density represents the potential to extract

power over a given area based on an estimate of the land area occupied by turbines and their combined generation capacity [23]. It does not account for variations in capacity factor, curtailment, operational availability, or wake effects.

In reviewing the literature, it is apparent that capacity density is frequently used interchangeably with "power density" though they can reflect different spatial quantities and are often expressed in different units [21,29]. To aid in interpretation of our work, here we draw a clear distinction between our focus on capacity density and the similarly termed "wind power density", which is a normalized measure of the accessible kinetic wind energy flux (W/m$^2$) [30]. The critical difference is that capacity density refers to the nameplate installation capacity that may be attained over a horizontally-defined ground area (i.e., how many megawatts worth of turbines can potentially be installed within one square kilometer of land?). In contrast, wind power density is a measure of the strength of the wind resource and describes the specific power flow within a vertical, cross-sectional domain (i.e., how much wind power can be generated per square meter of rotor swept area?). In addition, wind power density has been studied more intensely in the literature compared to capacity density. Numerous methods exist for quantifying wind power density (e.g., Mohammadi et al. [31]) whereas there are no established techniques for spatial modeling of capacity density as far as we are aware. We caution the reader that ambiguity and inconsistency in the usage of these terms among disciplines complicates interpretation and comparison across studies.

### 2.2. Role of Capacity Density in Technical Potential Modeling

Capacity density is a core component within the technical potential modeling domain alongside wind resource, system performance, and siting constraints [27] (Figure 1). In a traditional technical potential assessment, capacity density is used to estimate the amount of potentially installable capacity (in megawatts) and scales the estimates of turbine performance (i.e., capacity factors) to represent potential annual generation within a given areal extent [32]. Capacity density imposes operationally defined limits to areal energy production capacity and establishes a baseline for evaluating capacity expansion scenarios for future wind deployment [33].

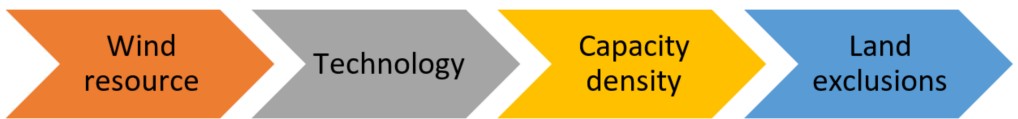

**Figure 1.** Major components of wind power technical potential. Turbine technology specifications determine the amount of energy that may be captured from the local wind resource. Capacity density establishes the maximum nameplate power capacity of turbines that is installable over a given area and reflects turbine spacing and siting requirements. In combination with spatial data on non-developable areas (i.e., land exclusions), these components are used to assess technical potential.

The annualized technical potential for each spatial unit ($k$) is computed using the following formula:

$$E_k = (A_k \delta_k)\eta_{a_k}\eta_{ar_k}CF_k \cdot 8,760h \qquad (1)$$

where $E_k$ is the technical potential for wind energy (MWh/year), $A_k$ is available land area (km$^2$), $\delta_k$ is the capacity density (MW/km$^2$), $\eta_{a_k}$ is the availability factor, $\eta_{ar_k}$ is the array efficiency factor, and $CF_k$ is the localized capacity factor for a selected turbine and hub height. Values used for availability factor ($\eta_{ar_k}$) and array efficiency ($\eta_{ar_k}$) may reflect long-term averages estimated using multiple years of generation data or may correspond to a typical year. 8760 is the generalized total annual hours (h) of operation. Although we express $\delta_k$, $\eta_{a_k}$, $\eta_{ar_k}$ using the $k$ sub-index to reflect that these quantities are spatially variable, in practice, they are conventionally considered to be homogeneous across broad geographic extents, as discussed further below. Our goal in this paper is to

illuminate spatial variation of capacity density ($\delta_k$) across the United States, which has critical downstream implications for estimation of wind power technical potential ($E_k$).

### 2.3. Approaches to Representing Capacity Density

The traditional approach to representing capacity density for wind technical potential applications has been one of simplification—assigning a uniform value across broad geographic regions [27]. Within this paradigm, methods for inferring capacity density generally fall into two categories (Table 1). A common method is to estimate the average capacity density using observations of operational wind farms that include their installed capacity and associated land area (i.e., project footprint). Though this has the benefit of grounding estimates of technical potential based on observed densities, by reducing wide-ranging observations to an average value, this type of characterization fails to persist the geographic variation associated with the underlying observations. Downstream effects of the averaging method on modeling technical potential can be significant because of the considerable geographic variation in observed capacity densities that exists at national scales [22,23,34]. Considering the diverse sociopolitical, physiographical, and environmental factors with influence over the proceedings of wind farm development, it should not be surprising that wind farm configurations and associated land use are complex [35–37]. Still, it is common practice for findings from limited observational studies to be applied to national or global extents despite these generalizations being based on data obtained from individual wind projects and lacking broader geographic representation.

A second method assumes that capacity densities can be estimated exclusively as a function of turbine technology. Using a turbine model's rotor diameter, general rules of thumb regarding spacing of turbines with and orthogonal to prevailing wind direction are applied to estimate required land area per megawatt. Although each method for estimating capacity density has its own set of tradeoffs to approximate real-world installation patterns and representativeness of turbine specific spacing requirements, both are typically applied in ways that ignore geographic differences in potential installed capacity. In some cases, land suitability models have been used to partition the landscape into areas amenable for wind development [38–40], though this approach is independent of any consideration of geographic variation in capacity density.

Capture of this geographic variation in potential installed capacity has remained elusive in published studies where the application of uniform capacity densities is established as standard practice (e.g., [27,38,41–44]). Considerable uncertainties in the assessment of technical potential can arise from assumptions about capacity density. In an in-depth analysis of global land-based wind energy potential and its sensitivity to input parameters, Zhou et al. [36] found that capacity density had a significant impact on wind energy potential with global estimates (petawatt-hours) varying between $-60\%$ and $+80\%$ of a reference scenario (5 MW/km$^2$) based on the magnitude of assumed capacity density (2–9 MW/km$^2$). In a more recent study analyzing wind economic potential within the context of urbanization, Herran et al. [45] found capacity density was the single-most important factor affecting potential aside from economic parameters.

**Table 1.** Examples of conventional approaches to inferring spatially-uniform estimates of capacity density for wind technical potential modeling applications.

| Approach for Representing Capacity Density | Description | Application | Assumed Capacity Density [1] (MW/km$^2$) | Reference |
|---|---|---|---|---|
| *Average wind turbine installation density methodology* | Relies on empirically-derived estimates of capacity density based on project-level information on wind farm footprint and installed capacity. Resulting capacity densities are considered to be spatially uniform (i.e., they do not vary geographically). | U.S. | 5 | [27] |
| | | Global; land suitability factors combined with fixed capacity density to adjust local geographic potential | 4 | [39] |
| | | Global | 5; range of 2–9 MW/km$^2$ evaluated for sensitivity analysis | [36,45] |
| | | U.S. | 3 | [13] |
| | | India | 9 | [26] |
| | | Europe U.S. China | 19.8 21.7 48 | [16] [2] |
| *Turbine spacing methodology* | Estimates turbine-specific capacity density using rule of thumb minimum spacing requirements based on rotor diameter (D). Resulting densities are implemented as spatially uniform. | Global; turbine spacing computed using 4D × 7D | 9 | [38] |
| | | Finland; turbine spacing computed using 5D × 7D | 5.3–10.6 | [44] |
| | | Global; turbine spacing computed using 4D × 7D | 8.9 | [43] |
| | | Global; turbine spacing computed using 5D × 10D; local geographic potential adjusted following Hoogwijk et al. [39] | 6.5 | [40] |

**Table 1.** *Cont.*

| Approach for Representing Capacity Density | Description | Application | Assumed Capacity Density [1] (MW/km²) | Reference |
|---|---|---|---|---|
| | | Europe; turbine spacing computed using 4.375D × 4.375D based on Enevoldsen and Valentine [46] | 11.1 | [19] |
| | | Saudi Arabia; turbine spacing computed using 5D × 7D | 4.9–7.9 | [47] |

[1] In cases where authors only reported turbine spacing we converted these requirements to capacity density to facilitate comparison among studies. We used information on turbine spacing and rotor diameter to determine area requirements and used reported turbine capacity to quantify capacity density as outlined in the Materials and Methods Section. [2] We present the mean capacity density reported by the authors for a subset of the geographies assessed. They recommend using the mean density as an appropriate reference when estimating the upper bounds of achievable capacity installation using their calculations.

### 2.4. Spatial Drivers of Capacity Density

Developing generalizable intuitions about capacity density is challenging because it is highly variable across sites and because multiple spatial factors influence wind farm configurations, which drive land requirements [39]. From an engineering perspective, capacity density reflects design principles that seek to maximize array efficiency and energy production while minimizing wake losses to downwind turbines. In reality, however, wind projects are subject to siting regulations, require negotiation of turbine placement with land owners, and are otherwise constrained by additional criteria, including minimization of construction costs. In addition, wind farm configurations are known to differ by land cover type [23], implicating a regional component in capacity density variation. As observed by Denholm et al. [23], adherence to these practical considerations creates project configurations that differ from what may be expected via theoretical approaches.

Diffendorfer and Compton [48] conducted a detailed geospatial analysis of land transformation due to wind development. Though they did not explicitly focus on capacity density, their findings have important implications for wind potential. In addition to discovering that land cover and topographic variables accounted for observed differences in the magnitude of land transformation across wind farms, they found that turbine spacing varied by landscape setting. Consistent with findings reported by Denholm et al. [23], Diffendorfer and Compton [48] noted that wind farms in agricultural areas exhibited larger distances between turbines than those in untilled landscapes, with spacing becoming progressively tighter in forest, grassland, hay, and shrub cover types. These patterns in agricultural lands, they hypothesized, were produced as a result of lease agreements with landowners and also reflected setbacks due to zoning regulations. Holding constant all other factors, increased turbine spacing leads to decreased capacity density. These findings shed light on some of the plausible mechanisms driving geographic variation in capacity density and further highlight the need to incorporate this information into energy planning.

### 3. Materials and Methods

In the following section, we describe our quantitative approach to developing spatially-explicit predictions of capacity density. We first establish a comprehensive collection of observed capacity densities for operational onshore wind farms in the United States. Using this set of information, we then train a machine learning algorithm to predict observed capacity densities based on a curated set of geospatial variables and validate the model against independent test data. We generate spatially-explicit predictions of capacity density for the United States. Lastly, we use model interrogation techniques to interpret mechanisms driving model behavior and compare the predictions from our validated model against a leading capacity density benchmark frequently used in U.S. wind power

technical potential assessments. A high-level visual overview of this workflow is provided in Figure 2. We explain each of these steps in more detail in the subsequent sections.

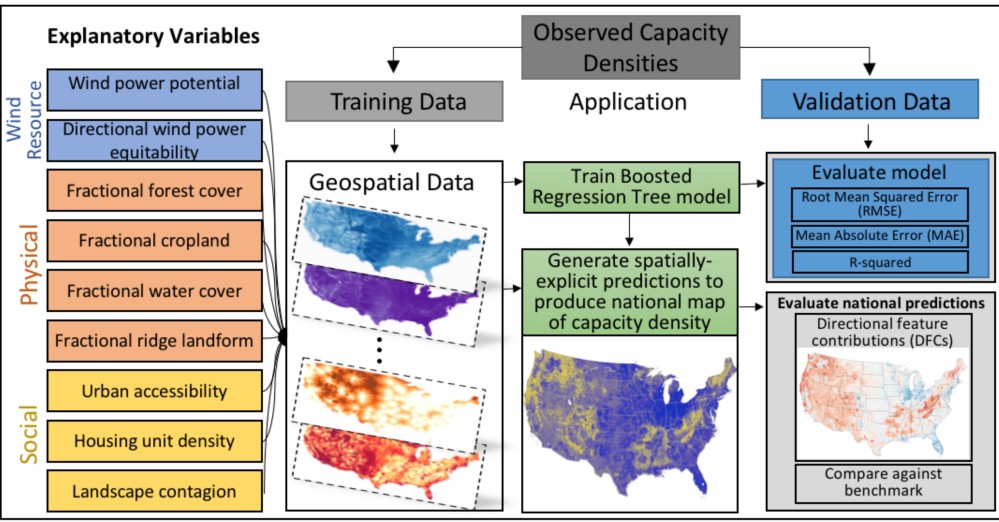

**Figure 2.** Schematic showing the data-driven workflow used to generate spatially-explicit predictions of capacity density for the United States (U.S.). We compile observed capacity densities for a comprehensive set of operational wind farms in the U.S. and associate those with geospatial variables characterizing physical, social, and environmental drivers that explain geographic variation in capacity density. The boosted regression tree model is trained using a subset of the observations and validated against independent test data to assess its ability to generalize to other locations. We use the validated model to generate a national map of capacity density and apply evaluation techniques to interpret and compare our predictions against a standard benchmark that assumes spatially-uniform capacity densities. Workflow visualization inspired by Lee et al. [49].

### 3.1. Wind Farm Data

We obtained spatial data on wind turbine installations from the U.S. Wind Turbine Database (USWTDB; [50]), which provides a comprehensive snapshot of land-based and offshore wind turbines that accounts for more than 58,000 operational turbines. Spatial coordinates along with project information and turbine specifications are provided for each record. We grouped turbines into distinct wind farms based on a unique project identifier. Project-level properties were computed for each farm describing the number of turbines, installed capacity, and turbine vintage. Wind farms were selected according to multiple criteria to ensure they would sufficiently represent contemporary wind deployment patterns. For the purposes of this analysis, we defined relevant wind farms as those land-based farms constructed in the conterminous United States (CONUS) after 2005 and having a minimum installed capacity of 20 MW [23]. We considered these to reflect contemporary utility-scale wind projects. Collectively, this set of farms represents 81,273 MW or 92.7% of total installed capacity in the United States. Multiphase farms are typically built incrementally over time, expanding outward from existing farms and sharing similar names with previous developments (e.g., Wind Farm I and Wind Farm II). We applied Levenshtein's distance algorithm [51], which scores sets of text strings based on a similarity measure, to identify multiphase farms based on their project name. We excluded all phases of multiphase farms to ensure that derived capacity densities would accurately depict total land requirements. Idiosyncrasies in phased development patterns could otherwise lead to erroneous estimates. In total, we excluded from this analysis 282 wind farms representing individual phases of development (28,047 MW). Doing so allowed us to focus on capturing cumulative as opposed to incremental development patterns.

### 3.2. Wind Farm Area Estimation

Quantification of wind farm area is necessary for evaluating capacity density but can be challenging because it requires defining spatial boundaries around turbines that lack an observable, comprehensive geographic footprint. Attempts to estimate wind farm land use generally fall into two categories that differ in terms of geographic scope: (1) twind farm area may be described narrowly in terms of direct impact area (i.e., surface area affected during turbine construction and experiencing ongoing impacts that are due to associated infrastructure) or (2) more broadly in terms of total project area that includes areas surrounding impacted zones and encompasses leased area when known [23,52]. Direct impact area is estimated to comprise 2–5% of total area on average based on the broad definition of project area. Here, we focused on total wind plant area as a relevant unit for understanding comprehensive space requirements of wind farms. Using this definition, we considered total area to encompass all lands occurring within the outermost perimeter of the turbine layout. Assuming a typical setback distance of 300 m [53], we constructed convex hull polygons around each farm's turbines to produce the minimum convex geometry containing all turbines. We computed wind farm area using the convex hull geometries. Note that these geometries do not explicitly account for non-turbine features associated with the plant (e.g., substation, operation, and maintenance building).

### 3.3. Capacity Density Characterization

Capacity density quantifies the attainable level of energy generating capacity per unit area. We defined wind farm capacity as the aggregate sum of installed turbine capacities. Specifically, we computed capacity density for each wind farm ($k$) as:

$$\delta_k = \left( \sum_{i=1}^{n} t\_cap_i \right) / A_k \tag{2}$$

where $\delta_k$ is the capacity density (MW/km$^2$) for farm ($k$), $t\_cap_i$ is the rated capacity (MW) of turbine ($i$), $n$ is the number of installed turbines at farm ($k$), and $A_k$ is the estimated wind farm area (km$^2$) at farm ($k$).

To manage the uncertainties associated with our methods for quantification of area for very small and linear wind farms, we excluded farms with capacity densities exceeding the 95th percentile of observed capacity densities (7.1 MW/km$^2$). Wind farms included in this analysis had a mean observed capacity density of 2.74 $\pm$ 1.40 MW/km$^2$.

### 3.4. Spatial Sampling of Wind Farm Characteristics

We considered capacity density as a wind farm-level attribute that can be explained by geospatial variables describing characteristics of the farm. In order to capture fine-grain geographic variation that exists among these variables, particularly for larger farms that may be spread out over significant distances and in heterogenous environments, we sampled geospatial characteristics at all turbine locations within each farm. This sampling approach produced a set of spatially replicated observations [54] linking observed farm-level capacity densities with spatial variables characterized for each turbine (Figure 3). Turbine-level samples may be considered non-independent at the farm-level, however. Thus, to minimize pseudoreplication that would result from the use of non-independent samples [55] in our modeling, we utilized a random subset of turbine samples from each farm. We controlled for the number of samples selected according to farm size, sampling in proportion to the square root of the number of turbine installations at each farm. This procedure ensured that larger farms had comparatively greater numbers of samples (with a sample's geospatial characteristic obtained at the turbine-level) than small farms based on the need for higher density sampling to adequately capture geographic variation occurring within the footprint of large wind farms.

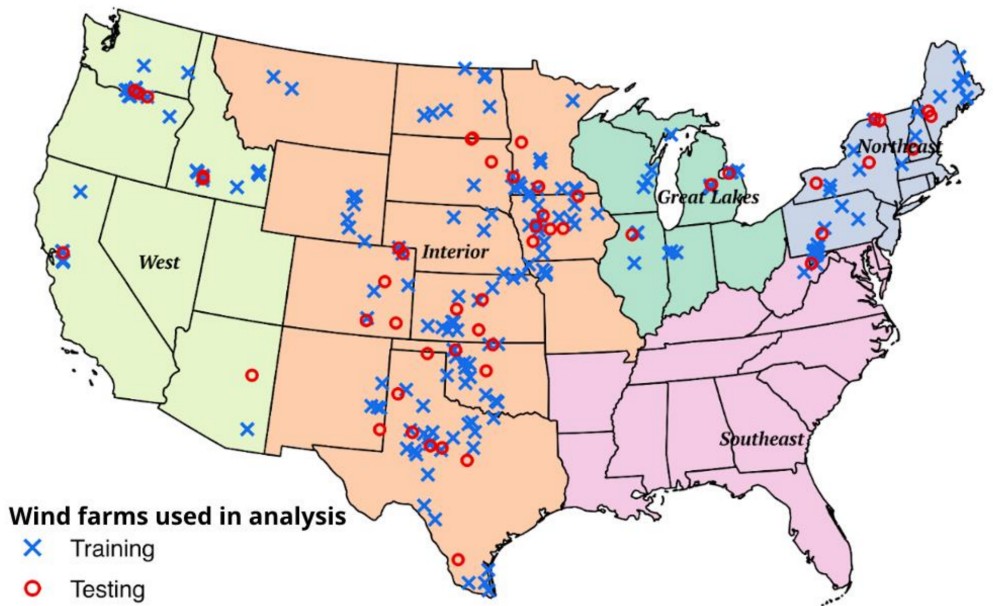

**Figure 3.** Spatial distribution of wind farm facilities used in model training and testing. Centroids of selected wind farms from the U.S. Wind Turbine Database [50] are displayed here in tandem with wind regions defined by the Department of Energy's Office of Energy Efficiency and Renewable Energy (DOE EERE) [2]. As shown here, training and testing datasets were each designed to include observations of wind farms located in all regions of the country. This selection was done to minimize geographic biases that could otherwise limit the ability of the model to generalize across regions.

### 3.5. Geospatial Data

Based on prior observation of wind farm layouts, we hypothesized that geospatial factors relating to wind resource, land cover, terrain, land use/ownership, and population density would influence capacity density. To develop predictive relationships with capacity density, we produced a suite of geospatial characterizations for a total of 19 variables related to these themes. All variables were characterized at 2 km resolution based on the grid of the wind resource [56]. A description of the candidate explanatory variables is presented in Table 2. Methodological details and additional metadata are provided in the Supplementary Materials. Explanatory variables were stacked to produce multiband images and sampled at each turbine location using Google Earth Engine [57]. This produced a train/test dataset containing capacity density and associated variables along with wind farm IDs.

**Table 2.** The candidate set of geospatial variables used to model capacity density. Descriptive statistics are provided for the conterminous United States (CONUS).

| Variable | Description | Minimum | Maximum | Mean | Standard Deviation |
|---|---|---|---|---|---|
| ACC * | Urban accessibility measured through travel time to nearest urban center (minutes); [58] | 0 | $9.45 \times 10^2$ | $1.21 \times 10^2$ | 93.2 |
| BUI | Built-up intensity of residential and commercial buildings (m$^2$); derived from [59] | 0 | $1.1 \times 10^7$ | $2.65 \times 10^3$ | $3.27 \times 10^4$ |

**Table 2.** *Cont.*

| Variable | Description | Minimum | Maximum | Mean | Standard Deviation |
|---|---|---|---|---|---|
| CLF | Fractional areal extent of cliff landform (unitless); derived from [60] | 0 | 38 | $4.91 \times 10^{-2}$ | 0.64 |
| CON * | Landscape metric describing contagion (i.e., the spatial "clumpiness" of unsuitable lands; wind exclusions); derived from [7] | 5 | $1.00 \times 10^2$ | 71.3 | 23.3 |
| CRG | Fraction of pixel containing ridge landform (unitless); derived from [60] | 0 | 1 | 32.1 | 0.47 |
| DIV | Fractional areal extent of divide landform (unitless); derived from [60] | 0 | 27 | $6.19 \times 10^{-2}$ | 0.72 |
| GAP | GAP 1&2 status protected lands (%); derived from [61] | 0 | 1 | $6.85 \times 10^{-2}$ | 0.023 |
| HUD * | Housing unit density (units/km$^2$) [62] | 0 | $1.58 \times 10^4$ | 17.2 | $1.08 \times 10^2$ |
| LBL | Wind regions defined by Lawrence Berkeley National Laboratory [2] | N/A | N/A | N/A | N/A |
| LC1 * | Fractional areal extent of water land cover class (unitless); derived from [63] | 0 | $1.00 \times 10^2$ | 1.84 | 9.98 |
| LC2 | Fractional areal extent of developed land cover class (unitless); derived from [63] | 0 | $1.00 \times 10^2$ | 5.45 | 12.9 |
| LC3 | Fractional areal extent of barren land cover class (unitless); derived from [63] | 0 | $1.00 \times 10^2$ | 1.16 | 7.67 |
| LC4 * | Fractional areal extent of forest land cover class (unitless); derived from [63] | 0 | $1.00 \times 10^2$ | 24.9 | 31.2 |

**Table 2.** *Cont.*

| Variable | Description | Minimum | Maximum | Mean | Standard Deviation |
|---|---|---|---|---|---|
| LC5 | Fractional areal extent of shrubland cover class (unitless); derived from [63] | 0 | $1.00 \times 10^2$ | 22.1 | 33.0 |
| LC7 | Fractional areal extent of herbaceous land cover class (unitless); derived from [63] | 0 | $1.00 \times 10^2$ | 14.7 | 25.7 |
| LC8 * | Fractional areal extent of planted/cultivated land cover class (unitless); derived from [63] | 0 | $1.00 \times 10^2$ | 22.7 | 31.2 |
| LC9 | Fractional areal extent of wetlands land cover class (unitless); derived from [63] | 0 | $1.00 \times 10^2$ | 5.02 | 13.8 |
| LFR | Landform regions; derived from [64] | N/A | N/A | N/A | N/A |
| LSP | Fractional areal extent of lower slope landform (unitless); derived from [60] | 0 | $1.00 \times 10^2$ | 39.3 | 15.7 |
| LU1 | Land use class (level I); derived from [65] | N/A | N/A | N/A | N/A |
| MWS | Mean long-term wind speed at 80 m hub height (m/s); derived from [56] | 1.35 | 14 | 6.26 | 1.08 |
| POP | Population density (persons/km$^2$); derived from [66] | 0 | $3.18 \times 10^4$ | 41.3 | $2.58 \times 10^2$ |
| RDG * | Fractional areal extent of ridge landform (unitless); derived from [60] | 0 | 36 | 1.59 | 3.77 |
| RIX | Fractional areal extent exceeding critical slope threshold (unitless); derived from [67] | 0 | 1 | $6.30 \cdot 10^{-2}$ | 0.17 |
| SLF | Fractional areal extent of suitable landforms (i.e., not cliff or valley; unitless); derived from [60] | 1.90 | $1.00 \times 10^2$ | 86.2 | 10.8 |

**Table 2.** *Cont.*

| Variable | Description | Minimum | Maximum | Mean | Standard Deviation |
|---|---|---|---|---|---|
| TRC | Fractional areal extent of tree cover); derived from [60] | 0 | $1.00 \times 10^2$ | 26.0 | 28.2 |
| USP | Fractional areal extent of upper slope landform (unitless); derived from [60] | 0 | $1.00 \times 10^2$ | 43.3 | 15.2 |
| VLY | Fractional areal extent of valley landform (unitless); derived from [60] | 0 | 70 | 13.0 | 9.47 |
| WEQ * | Power equitability metric describing the evenness of wind energy contributions at 100 m hub height across compass directions; derived from [56] | 0 | 99 | 90.7 | 8.46 |
| WEX | Fractional areal extent of wind exclusions; derived from [7] | 1 | 1 | 1 | 0 |
| WND * | Dimensionless wind energy at 100 m hub height (unitless); derived from [56] | 0 | $1.94 \times 10^2$ | 96.4 | 25.0 |

An asterisk (*) indicates the variable was selected for use in the final predictive model after screening candidate variables based on correlation coefficient.

### 3.6. Machine Learning Using Boosted Regression Trees

Research applications of machine learning have increased dramatically within the last decade [68], supporting new insights through modeling. Although well known for their predictive abilities and success across disciplines, these methods have been criticized as black boxes, perceived to offer no insight into their inner workings and decisions. However, the advent of tools to explore model functionality have enabled advances in model interpretation [69,70], providing researchers with richer insight into machine learning models and expanding their utility.

In particular, boosted regression trees, a class of ensemble machine learning techniques that leverage collections of relatively simple models (i.e., weak learners [71,72]) have shown remarkable success on a wide variety of tasks without sacrificing interpretability [73,74]. Several desirable features of boosted tree models, commonly known as gradient-boosted models, have contributed to their popularity. These features include the ability to accommodate continuous and categorical explanatory variables, robustness and insensitivity to outliers, modeling of interactions, and tools for interpretation [75]. The boosting algorithm refers to a sequential tree-building process wherein trees are built in successive steps to explain model residuals produced at the previous stage. Contributions of each tree are limited by a learning rate hyperparameter to prevent overfitting that would otherwise occur. Other key hyperparameters that control elements of the learning process include number of

trees, maximum tree depth (i.e., interaction depth), and regularization parameters intended to penalize more complex models.

Model Building

We partitioned the data into separate training and test sets containing 80% and 20% of the records, respectively. In theory, a model trained on a subset of samples from a given wind farm could produce highly accurate predictions for another subset from the same farm while being a poorly performing model in other "unseen" contexts. To encourage generalization and prevent overfitting to known capacity densities, training and test datasets were produced in a stratified fashion according to wind farm IDs, thereby ensuring records for individual wind farms would appear exclusively in either the dataset used to develop the model or the dataset used to evaluate the trained model. We further discouraged artificial inflation of model performance estimates by sampling one record per farm for the test dataset. Combined, the training and test datasets contained 1233 samples from 190 and 50 unique farms respectively.

Hyperparameter tuning of machine learning models is a dataset-dependent task and is a necessary step toward model optimization. We created a tune grid defining the extent of our hyperparameter space focusing on number of trees (100–1000), maximum tree depth (3–14), and learning rate (0.001–0.050). To guard against the construction of overly complex trees, we enabled post-pruning using a tree complexity value of 0.01. We performed a randomized search of the tune grid, evaluating model performance for 200 unique hyperparameter combinations against a withheld portion of the validation data created using three-fold cross-validation. We selected the set of hyperparameter values that best minimized mean squared error (MSE) on the training dataset. We used correlation analysis to identify less relevant and redundant variables known to mask variable importance and hinder model performance in boosted regression trees. Variables with an absolute correlation > 0.4 were dropped, giving preference to those hypothesized to have a clearer or more direct linkage (i.e., mechanism) with wind plant capacity density. The nine explanatory variables included in the final model are identified in Table 2.

Using the boosted regression trees algorithm, we modeled observed capacity densities for wind farms contained in the training set using the geospatial predictor variables presented in Table 2. Next, we validated the trained model against the independent test set to characterize its ability to generalize to other wind farm locations. We used the validated model to generate localized predictions of capacity density at greater than two million locations across the United States to produce a national map. Lastly, we interpret the model predictions using model interrogation techniques and compare our findings with established benchmarks in the field of wind power technical potential modeling. For an overarching visual depiction of this analysis workflow, we refer the reader to Figure 2.

Modeling was conducted using the TensorFlow Boosted Trees (TFBT) implementation [76] built on the TensorFlow machine learning framework [77]. All analyses were performed using the Python programming language [78].

## 4. Results

In this section, we present an overview of our results. We begin with a rigorous validation assessment of our trained model against independent test data. We then use the model to present our spatially-explicit predictions of capacity density for the United States, providing additional interpretation to explain the mechanisms driving model behavior.

### 4.1. Validation

Hyperparameter settings and accuracy metrics for the best performing model are provided in Table 3. Against the independent test set, the boosted regression tree model achieved mean absolute error (MAE) of 1.02 MW/km$^2$ and a root mean squared error (RMSE) of 1.25 MW/km$^2$. As expected, the model performed better on the training set than on the test set, despite implementing regularization techniques (e.g., low learning rate and

post-pruning). Explanatory power measured against the test set produced an r-squared ($R^2$) of 0.40, which is a more robust characterization of how well the model generalizes.

**Table 3.** Results for the best model including model configuration as determined via hyperparameter optimization and accuracy metrics obtained against an independent test set. Accuracy metrics are mean absolute error (MAE), root mean squared error (RMSE), and r-squared ($R^2$).

| Number of Trees | Maximum Depth | Learning Rate | MAE | RMSE | $R^2$ |
|---|---|---|---|---|---|
| 200 | 10 | 0.005 | 1.02 MW/km$^2$ | 1.25 MW/km$^2$ | 0.40 |

Inspection of the residuals indicates that model error is centered on zero and that predictions lack bias according to the residual distributions (Figure 4). In addition, test set residuals are more uniformly distributed and show that the model does not tend to systematically over- or underpredict capacity density. Although we found a stronger fit on the training set according to the $R^2$ metric (0.72), we note that the distribution of residual error observed in these samples indicates that our implementation of regularization techniques (low learning rate and post-pruning) were effective.

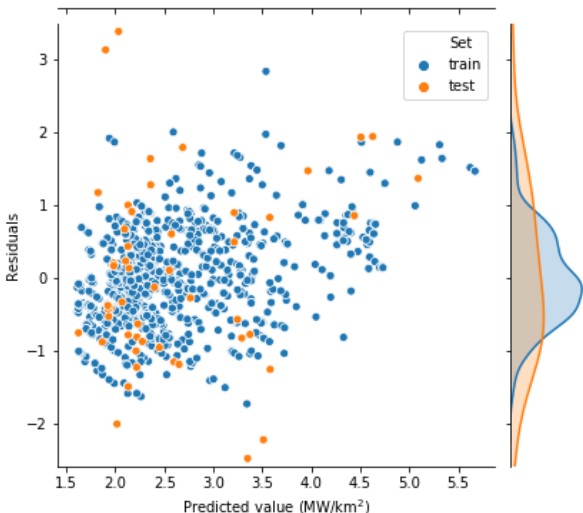

**Figure 4.** Residual plot showing prediction errors for train and test sets. The accompanying density plot (right-hand side) shows that the distribution of residuals centers around zero for both sets used in model evaluation.

*4.2. Model Interpretation*

Because not all model variables contribute equally, we gauged their contributions to the model using a feature importance method established by Friedman [79]. Feature importance values indicate the relative importance of a variable within a boosted tree model and are scaled such that they sum to one when aggregated across all variables. A feature's importance is characterized by measuring its improvements to the model averaged over all trees. Features with greater importance values tend to be more influential. Figure 5 depicts global feature importance in the model indicating that directional wind power equitability (WEQ), urban accessibility (ACC), forested land cover (LC4), and dimensionless wind energy (WND) have the greatest overall contributions, according to this measure (see Table 2 for variable descriptions). Extent of ridge landforms (RDG), housing density (HUD), and extent of planted/cultivated land (LC8) expressed moderate feature importance values, while extent of water cover (LC1) had the least importance.

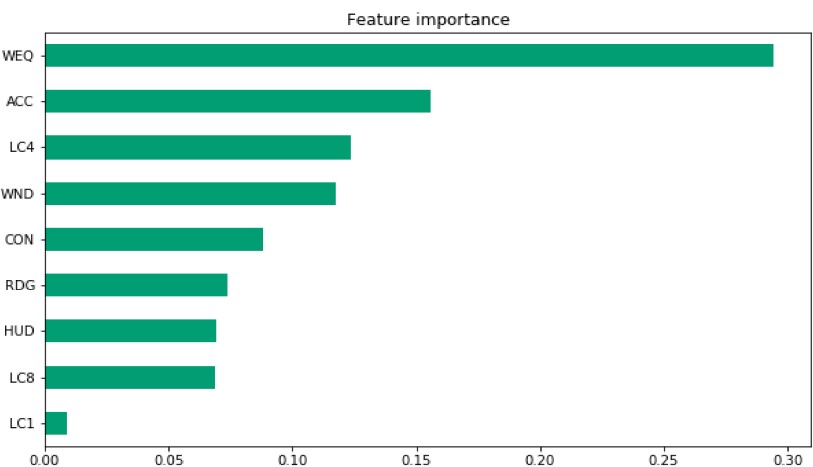

**Figure 5.** Feature importance values for the selected set of explanatory variables (Table 2), showing contributions to the boosted regression tree model. These values collectively sum to one across model variables. In order of decreasing feature importance, the top four variables are directional wind power equitability (WEQ), urban accessibility (ACC), forested land cover (LC4), and dimensionless wind energy (WND).

To expand our understanding of model behavior, we employed a technique pioneered by Palczewska et al. [69] to gain more granular insight into the basis for individual predictions. Directional feature contributions (DFCs) describe the nature of a variable's effect on the outcome and illuminate how different variables affect individual predictions. To compute DFCs, each prediction instance is run through the ensemble of trees, traversing the appropriate decision path and attributing changes in the predicted value to the variable included at each node split [80]. This technique complements the feature importance assessment and affords deeper insight into model behavior by enabling analysts to scrutinize how various model features (i.e., variables) contribute to specific predictions. Contributions indicate the magnitude and direction of influence for feature variables on the predicted value for capacity density. Contributions are standardized relative to the distribution of capacity density observed in the training set such that a contribution of zero indicates that a feature has not shifted the predicted capacity density value from the mean observed value. Positive contributions indicate that the feature value increased the prediction relative to the mean observed capacity density and negative contributions indicate the opposite. An intuitive property of DFCs is that for any particular example the sum of these contributions plus the bias term (i.e., mean value of capacity density from the training set) equals the predicted value. Aggregation of DFCs across a representative set of examples (i.e., a model evaluation set) to produce a distribution of contributions yields a global depiction of these effects [81].

Figure 6 shows DFCs for a prediction made for a sample wind farm from the test set. Deviations from zero (i.e., the mean capacity density from the training set) indicate a feature has had a positive effect (increase) or negative effect (decrease), respectively, on the predicted value relative to this mean. For this particular example, WND and HUD had the greatest effect on capacity density, both causing an increase in the predicted value. ACC and LC4 decreased the capacity density prediction, though these contributions were minimal in this instance. DFCs for this prediction case are shown alongside the global distributions to illustrate that feature contributions vary among samples. These distributions are generated in reference to predictions made for samples from the test set. The distribution of WND contributions are centered on zero and have a long right tail indicating that capacity density predictions for some samples were greatly increased (upwards of 0.5 MW/km$^2$) due to contributions by the WND variable. In contrast, contributions of LC4 tended to be negative, indicating that the forest cover variable largely had the effect of reducing predicted values of capacity density for test set instances. For the samples contained in the test set, the urban accessibility (ACC) variable

tended to decrease capacity density predictions though these contributions were variable as indicated by the high variance evident in the global distribution.

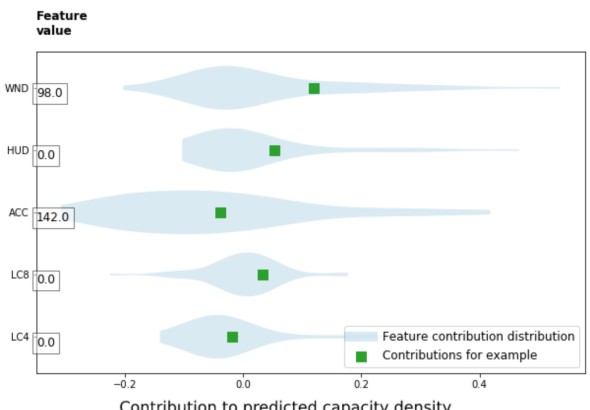

**Figure 6.** Feature values (left side) and contributions for the top five features for an example wind farm sampled from the test set. Variables are described in Table 2. Contributions indicate the magnitude and direction of influence for feature variables on the predicted value for capacity density. Contributions are described relative to the mean capacity density value from the training set, with positive contributions indicating that the feature value increased the prediction and negative contributions indicating the opposite. Distributions describe feature contributions taken from all samples from the testing set.

### 4.3. Spatially-Explicit Predictions of Capacity Density

A map of predicted capacity density for CONUS is shown in Figure 7. Including all lands, we found a mean predicted national capacity density of $2.82 \pm 0.75$ MW/km$^2$ with values ranging between 1.6 and 5.7 MW/km$^2$. Findings were similar when a standard set of land-based exclusions [27] were used to exclude likely undevelopable areas, with a mean predicted capacity density of $2.76 \pm 0.72$ MW/km$^2$.

The following results are representative of the non-excluded land area of CONUS (i.e., are deemed suitable for wind deployment as described above). Investigation of the cumulative distribution of capacity densities revealed differences among wind regions (Figure 8). We found predicted capacity densities to be generally lowest in the Great Lakes region and highest in the West, with median values of 1.9 and 3.0 MW/km$^2$, respectively. At 2.8 MW/km$^2$, the Northeast had the second-highest capacity density as measured using the median predicted value. Interestingly, we found similar median capacity densities for the interior of the country (2.6 MW/km$^2$) and the Southeast (2.5 MW/km$^2$). Areas of low capacity density (<2.0 MW/km$^2$) were present in all regions as were areas with high capacity density (>5.0 MW/km$^2$), with the exception of the Great Lakes, where the maximum capacity density was 4.8 MW/km$^2$.

### 4.4. Mechanisms Driving Capacity Density Predictions

At the national extent, intraregional and interregional variability in capacity density are apparent, reflecting the influence of regional- and local-scale drivers. We mapped our CONUS predictions to their corresponding directional feature contributions to further aid in model interpretation and to provide spatial contextualization of the mechanisms driving model behavior. Figure 9 uses the concept of DFCs introduced earlier to show how key variables differentially contribute to capacity density predictions in terms of both magnitude and direction of effect. These interpretations deconstruct grid-level predictions to explain them on the basis of individual variables. As described above, DFC values are presented relative to the mean capacity density observed in the training set, so a negative value for a location indicates that a variable within that sample (i.e., location) has contributed to a predicted capacity density value less than the observed mean. Positive values indicate the opposite. Large-scale drivers include regionally varying wind resource

as characterized by wind rose equitability (WEQ) and potential energy (WND). Inspection of the DFC maps show that WEQ tends to decrease capacity density predictions across large sections of the Midwest, Southeast, and coastal portions of the Northeast. These areas correspond with high wind rose equitability, which indicates that sites whose potential energy is evenly distributed across multiple compass directions (i.e., lack a predominant wind direction) are associated with lower capacity densities, holding constant all other factors. In contrast, the presence of strong unidirectional wind resources leads to increased capacity density predictions. Although speculative, we hypothesize that this relationship may reflect how wind farms are configured to maximally capture wind resources with more concentrated spacing and thus higher capacity densities expected at sites where wind energy flows in a predominant direction.

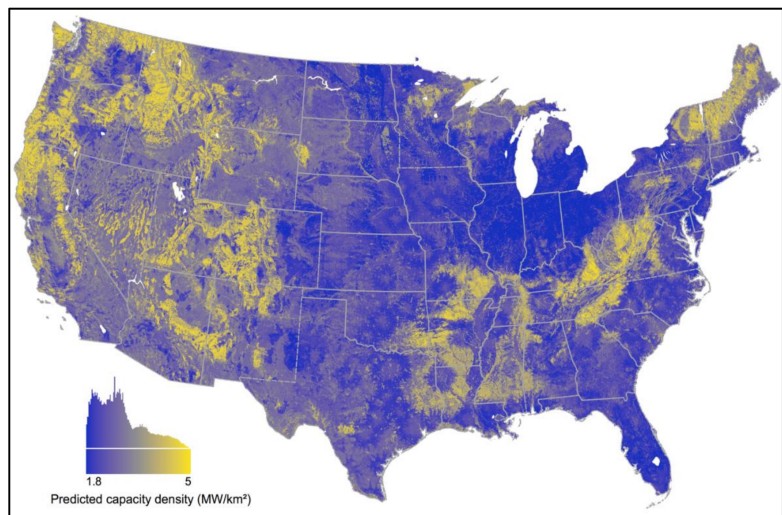

**Figure 7.** Map of predicted capacity density for CONUS. Areal capacity density represents potential nameplate capacity installation per unit ground area and is measured in $MW/km^2$. Predictions are made at a 2 km $\times$ 2 km spatial resolution based on the Wind Integration National Dataset (WIND) Toolkit wind resource dataset [56]. As described in previous sections, variations in capacity density reflect that wind farm land requirements are driven by multiple variables and that these phenomena contribute to substantial differences in both observed and predicted capacity densities across the country. Information on capacity density is needed to quantify wind deployment potential and is a key input into technical potential analysis and other downstream assessments (e.g., supply curve modeling). The histogram shows the distribution of predicted capacity density at the national scale.

The DFC maps reveal a more nuanced relationship between capacity density and wind energy potential. For example, high wind energy potentials found in the country's wind belt were associated with both neutral and slightly negative contributions to capacity density predictions. On the other hand, wind potential was found to have a positive contribution to capacity density throughout central Appalachia, the northernmost reaches of the Northeast, and the West, with the exception of valleys characterized by low wind resource. The lack of a clear association here may indicate an interaction of wind potential and other variables. Although the presence of a strong interaction could potentially confound a univariate interpretation, an inquiry into these secondary effects was beyond the scope of this analysis.

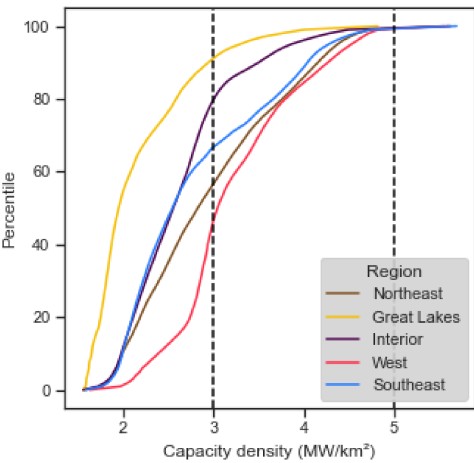

**Figure 8.** Cumulative distribution of predicted capacity density for major wind regions over non-excluded land areas. Dashed vertical lines refer to uniform capacity densities applied for estimates of U.S. wind power technical potential (3 MW/km$^2$: Mai et al. [13]; 5 MW/km$^2$: Lopez et al. [27]). See Lopez et al. [27] for definition of standard wind exclusions.

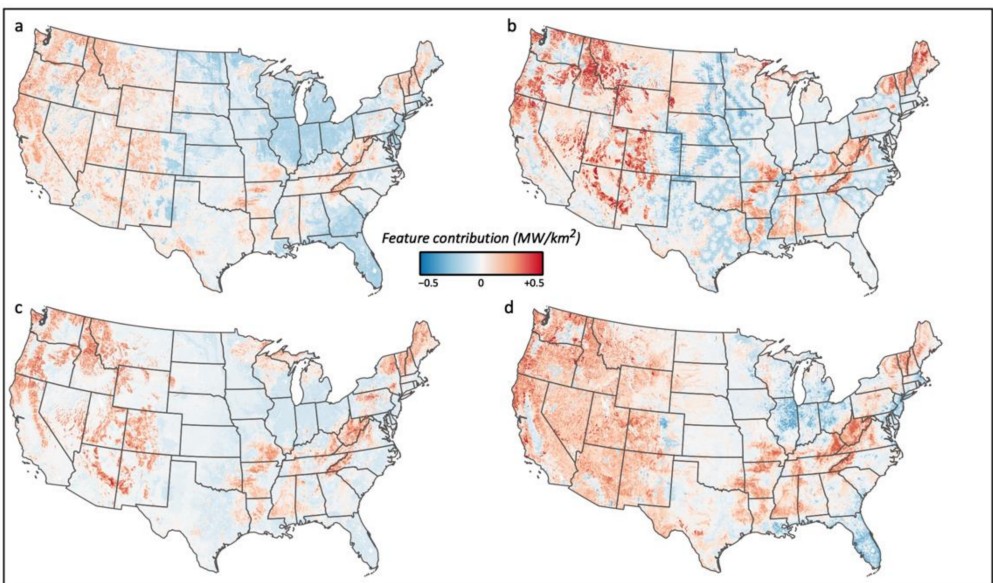

**Figure 9.** Directional feature contributions (DFCs) explaining the extent to which grid-level (2 km $\times$ 2 km) capacity density predictions for CONUS are attributable to the four most influential variables according to a feature importance measure: (**a**) wind energy equitability (WEQ), (**b**) urban accessibility (ACC), (**c**) forest cover (LC4), and (**d**) wind energy (WND). DFCs quantify the contribution of individual variables to each prediction and provide a local interpretation of model behavior. Intuitively, individual predictions are the sum of DFCs for all model variables plus the bias (i.e., mean capacity density value from the training set). DFCs are evaluated at a 2 km $\times$ 2 km spatial resolution based on the WIND Toolkit wind resource dataset [56].

Distinct patterns in the capacity density surface are also attributable to variation in urban accessibility (ACC), an indicator of remoteness as measured by travel time to the nearest city [58]. Here, the extent of a city is defined based on either the contiguous area exceeding a density of 1,500 people per square kilometer or a region characterized by predominantly built-up land cover and adjacent to a population source that exceeds 50,000 people. These patterns are expressed as concentric rings extending from urban centers and reflect significant regional differences in remoteness. Interestingly, the model tends to predict higher capacity densities for areas characterized by low (i.e., remote)

to moderate levels of accessibility. While the mechanisms behind this phenomenon are unclear, we speculate that it could be related to the presence of various land use constraints in moderately developed landscapes that lead to more confined turbine layouts and thus greater capacity densities. The ring of higher capacity density around Shreveport, LA, in Figure 8 is an example of how low urban accessibility bounded spatially by mostly forested area can result in distinct patterns of capacity density. Figure 9 shows how this pattern is explained in the model via urban accessibility, forest cover, and regional patterns of the wind resource. Regions with greater accessibility are likely to have more urbanization-related constraints (e.g., residences and roads). Accounting for these factors in working landscapes could require irregular turbine placement, effectively increasing wind farm areas and decreasing capacity densities accordingly.

In addition to these influences, we found high-density forest cover to be largely associated with increased capacity density predictions. This relationship is evident in the Northeast and Pacific Northwest and is also present in major forested mountain regions, including the Sierra Nevada, Rocky Mountains, and Appalachian Mountains. These findings are consistent with prior observations that wind farm layouts vary by land cover type [23] and that wind turbines tend to be more closely spaced in forested landscapes than in open (e.g., agricultural) landscapes [48]. As noted earlier, although multiple factors could contribute to why turbine spacing tends to be greater in agricultural settings, Diffendorfer and Compton [48] hypothesize that these patterns may be driven by the presence of land lease agreements and zoning restrictions that fragment developable areas, resulting in disjoint siting of turbines.

Considering all mechanisms that drive capacity density predictions, we find that regions with complex topography possessing robust wind resource and unimodal wind energy directionality combined with moderate forest cover and remoteness (i.e., are less accessible) appear to have the greatest capacity densities when generalized at the national level. In addition to these primary drivers, we attribute finer-grain variability in predicted capacity density to localized features that include terrain (RDG) and spatial development patterns as captured by housing density (HUD) and fragmentation of lands deemed suitable for wind deployment (CON).

## 5. Discussion

### 5.1. Comparison with Other Findings

To the best of our knowledge, this is the first map of capacity density that has been produced for a large geographic region such as the conterminous United States. Moreover, we are not aware of any other studies that have explicitly predicted localized capacity density at this scale. Conventional approaches applied to similarly large geographic areas and reported in Table 1 either assign geographic averages based on wind farm observations or estimate capacity density based on generic turbine spacing requirements. In addition, layout optimization analyses tend to be narrowly defined at the project level in terms of turbine technology, operations, and siting criteria and would be a poor basis for comparison. We therefore lack alternative sources from which to make a direct comparison against our model or predicted capacity density surface. Still, we find that the range of predicted capacity densities is generally in line with previously reported numbers. Among developable lands we found a mean predicted capacity of 2.76 MW/km$^2$ which is similar to the 3 MW/km$^2$ used by Mai et al. [13] and Lopez et al. [7], but substantially less than the 5 MW/km$^2$ used by Lopez et al. [27] for United States specific technical potential assessments. Nevertheless, as we have demonstrated, the spatially varying aspect of capacity density is a critical feature for the assessment of wind technical potential that is persistently overlooked. In assessing our findings in aggregate, we caution that a focus on national averages obscures important regional and sub-regional variation that is captured by our approach. Geographic limitations of conventional representations of capacity density are visualized in Figure 10, which highlights key regional deviations in predicted capacity density relative to a uniform benchmark.

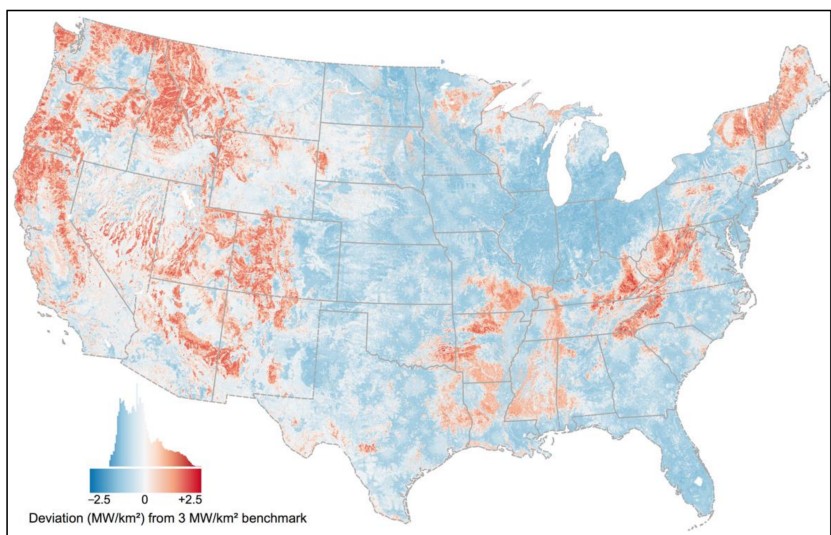

**Figure 10.** Differences between our geospatial predictions of capacity density and the 3 MW/km$^2$ benchmark used in recent national studies of wind power technical potential (e.g., Lopez et al. [7]; Mai et al. [13]). Deviations (MW/km$^2$) shown in the map illuminate where gross average estimates of capacity density applied uniformly at the national scale fail to capture local and regional variation with downstream implications for technical potential. Negative deviations indicate that our model predictions are less than the 3 MW/km$^2$ benchmark whereas positive deviations indicate the opposite. The histogram shows the national distribution of these deviations.

*5.2. Model Evaluation*

Evaluation metrics indicate that the model achieved suitable generalizability against the independent test set (MAE = 1.02 MW/km$^2$, RMSE = 1.25 MW/km$^2$). In particular, we believe that the prediction surface provides a useful depiction of spatially varying capacity density and is an improvement over existing approaches that assume a uniform capacity density across geographic space. Because capacity density is effectively used to scale the magnitude of available wind capacity and potential generation in assessments of technical potential [32], our achieved level of accuracy is suitable for supporting those types of inferences. Given the complexity of factors and decision-making processes that influence siting, design, and the ultimate success of wind energy projects, we believe that the observed performance (R$^2$ = 0.40) constitutes valuable explanatory power for real world data.

Moreover, we contend that while site-specific predictions have room for improvement, the regional trends identified in Figure 7 are meaningful and provide insight to the wind and general energy modeling community. Recognizing that other factors outside the scope of our analysis affect capacity density, we would expect a portion of the variance in capacity density to remain unexplained. These findings are consistent with our expectations of a meaningful and credible model yet allow for continued improvement in the future with the incorporation of finer-grain explanatory datasets and increases in training data afforded by additional wind farm deployment. For example, we hypothesize that many of these factors are at least, in part, social in nature and that bringing them to bear on future wind energy research will require sophisticated geospatial analysis of fine-grained ownership, zoning, and development data. Cumulative deployment effects may also impact regional capacity densities over time due to potential wind resource, social, and ecological considerations. Each of these are active areas of research that we anticipate will help cultivate more-precise geospatial insights of wind technical potential as it relates to additional spatial phenomena moving forward.

The ability to draw inference from the boosted regression tree model to implicate those factors found to be most important in explaining observed variation in capacity density is integral to formulating insights about key drivers and the nature of their association with capacity density. Employing model interrogation techniques, we determined that the

most important features were related to wind resource characteristics (WEQ and WND), accessibility to urban centers (ACC), and forest cover (LC4). Other relevant but less impactful features included the spatial patterning of lands considered unsuitable for wind development (CON), extent of ridge landforms (RDG), housing unit density (HUD), and extent of agricultural lands (LU8). DFC maps (Figure 9) provide additional insight into the predicted capacity density surface, explaining how these spatial variables contribute to model predictions.

These findings add to a focused and emerging body of evidence documenting the complexities of land area requirements of wind power. Although we demonstrate that geospatial factors shape the physical settings for wind farm development, they do so alongside sociopolitical, ecological, and engineering influences [39,48] that are difficult to capture at a national scale. In addition, we acknowledge that large-scale atmospheric processes control critical dynamics for the wind industry. For example, geophysical processes are known to limit the rate at which kinetic energy may be extracted from the atmosphere [82], setting the upper limit to hypothetically achievable capacity density [22]. Though from an operational perspective, the economic, social, and technical constraints are more likely to impose realized limits to wind potential [83].

Employing boosted regression trees as the modeling framework enabled the capture of these types of multidimensional relationships via their tree-based structure [75]. This is because models possessing deep trees with many splits (i.e., branches) effectively describe nuanced relationships that include variable interactions [73]. Although our current scope did not allow for investigation of variable interactions per se, we suspect that further assessment of the model would reveal key relationships within the interplay of topographic, land cover, wind resource, and social factors that affect wind energy potential. Additional research to crystallize these relationships would help further illuminate how regional patterns of deployment are affected by energy and zoning policies as well as other factors of interest to researchers and decision makers.

Development of meaningful machine learning applications requires that input training data are of high quality and are sufficiently representative to capture variation in the process being modeled. These design principles are reflected in our data curation and sample design. For example, to both support and evaluate the model's ability to generalize across regions, we trained and tested model performance using samples distributed throughout all regions of the country possessing operationally active commercial wind farms (Figure 3). Although we explicitly accounted for regional variation in model development and testing, we recognize that not all regions of the United States are represented by current wind farm deployment. However, to the extent that the associations between geospatial variables and capacity density captured by our model are translatable across geographies, we believe that spatial predictions for the Southeast are insightful for understanding relative regional differences in expected capacity density. Furthermore, we believe that the driving variables associated with wind resources, topography and other physical characteristics, and spatial development patterns are of universal relevance to wind farm deployment and are therefore likely to apply to other regions independent of their physical proximity to existing wind farms. Thus, rather than viewing the geographic distance between sites used for model development and prediction as relevant for inferring the model's ability to extrapolate, we see this challenge primarily as one that involves interpolation in variable as opposed to physical space.

## 6. Applications and Conclusions

The map of predicted capacity density presented here is representative of the spacing characteristics of current wind farm deployments and their associations with geospatial variables. As the wind industry continues to evolve [2], it will be important to monitor for changes in technology or geospatial deployment that could boost attainable capacity densities. For instance, wake steering technology innovations that enable higher turbine installation densities with mimimal impacts to energy production may be creating new

opportunities for wind deployment in land-constrained regions [84]. Absent fundamental shifts in the capacity density landscape, however, we believe that our derived capacity density layer provides a useful representation of wind potential for contemporary farms across the conterminous United States. Moreover, this machine learning approach is sufficiently generalizable to be extended to other countries where broad-scale planning and assessment of wind power potential is needed. Those efforts would benefit from the collection and curation of country-specific data on wind power deployment and geospatial characteristics and would enable deeper examination of spatial variation in potential capacity densities as demonstrated here.

Although future research will presumably improve upon our efforts, we believe that this approach reflects a step change in the research community's ability to characterize wind energy technical potential across broad geographic extents. Our hope is that making these data broadly accessible will support a more comprehensive research and planning process wherein insights about geospatial variation in wind energy potential are considered in future technology development and deployment scenarios.

*6.1. Applications*

Our approach to model geospatial variation in capacity density across broad geographic areas can improve how studies of technical potential inform local, regional, and national opportunities for future wind energy deployment. Advancing a geographically based understanding of differences in capacity density that shape technical potential is critically needed to improve the research and planning community's understanding of where, how much, and under what conditions wind energy might be deployed in the future. Characterization of spatial aspects of potential future wind plants is also critical for informing technology development that could mitigate social and ecological impacts from wind energy as it becomes integrated into the energy system of the future. Such knowledge is increasingly being used to guide long-term energy planning including for decarbonization scenarios and informs local, state, and federal policymakers as they consider legislation around clean energy adoption. To this end, technical potential is a principal input for the Integrated Planning Model (U.S. Environmental Protection Agency), the National Energy Modeling System (U.S. Energy Information Administration), and the Regional Energy Deployment System (used by the U.S. Department of Energy). These models provide enhanced insights when they directly reflect regional changes in available wind generation as a function of capacity density.

*6.2. Conclusions*

In conclusion, by demonstrating the potential variability of wind power capacity density across broad geographic extents, our spatially-explicit representation of capacity density better illuminates the potential land use implications of an expanding renewable energy portfolio [85,86]. Moreover, by using empirical data to predict potential characteristics of future deployments, the work informs the ways in which growing energy needs may be met [87] by wind deployment in diverse settings [45]. We contend that capturing this geographic context is essential to informed decision-making and long-term energy planning, especially as society grapples with a possible energy transition in the context of deep decarbonization objectives. In addition, by more precisely characterizing how wind energy might be deployed in regions with little prior development, the work can help inform potential new research and technology innovation needs including those that are specifically applicable to the types and sizes of wind power plants that are necessary to achieve further capacity additions across the conterminous United States. Notably, the regional differences in predicted capacity densities between the interior of the country and the Northeast suggest that innovations that are applicable to closely clustered turbines but more distributed plants will be more impactful in regions such as the Northeast while innovations focused on large plants with extended spacing between individual turbines will have greater impact in the interior.

New outcomes from our work also include an established methodology that provides a platform for continued advancement in geographic characterization of wind power technical potential. Additionally, our modeling has uncovered that factors beyond traditional wind energy considerations (e.g., wind technology and wind resource) including social characteristics of the landscape play a measurable role in affecting wind energy capacity density across space. Specific priorities for future research include increasing levels of resolution across national and continental scales, enriched characterization of social and ecological variables, and consideration of how cumulative effects of wind deployment might ultimately shape and influence wind energy technical potential. Fundamentally, we hope that our data-driven approach to characterizing spatial aspects of wind power systems will promote deeper investigations of wind technical potential that are grounded in geography and reflect the factors that explain geographic variation among operational wind farms.

**Supplementary Materials:** The following are available online at https://zenodo.org/record/4705111, Figure S1: Geographic variation in wind farm capacity density, Table S1: Geospatial data sets used in the modeling of capacity density, and supporting information on geospatial data and methods.

**Author Contributions:** Conceptualization, G.M. and D.H.-A.; methodology, D.H.-A.; validation, D.H.-A.; formal analysis, D.H.-A.; writing—original draft preparation, D.H.-A., G.M., and E.L.; writing—review and editing, D.H.-A., G.M., and E.L.; visualization, D.H.-A.; funding acquisition, G.M. and E.L. All authors have read and agreed to the published version of the manuscript.

**Funding:** This work was authored by the National Renewable Energy Laboratory, operated by Alliance for Sustainable Energy, LLC, for the U.S. Department of Energy (DOE) under Contract No. DE-AC36-08GO28308. Funding was provided by the U.S. Department of Energy's (DOE) Office of Energy Efficiency and Renewable Energy Wind Energy Technologies Office. The views expressed in the article do not necessarily represent the views of the DOE or the U.S. Government. The U.S. Government retains, and the publisher, by accepting the article for publication, acknowledges that the U.S. Government retains, a nonexclusive, paid-up, irrevocable, worldwide license to publish or reproduce the published form of this work, or allow others to do so, for U.S. Government purposes.

**Data Availability Statement:** The corresponding author will provide data at reasonable request.

**Acknowledgments:** We thank Anthony Lopez for his involvement with early stages of this work. He along with Ryan King provided constructive feedback that lead to invaluable improvements to the manuscript. In addition, we are appreciative of Ben Hoen for sharing his expertise on the U.S. Wind Turbine Database. A portion of the research was performed using computational resources sponsored by the Department of Energy's Office of Energy Efficiency and Renewable Energy and located at the National Renewable Energy Laboratory.

**Conflicts of Interest:** The authors declare no conflict of interest.

## Abbreviations

The following abbreviations are used in this manuscript:

| | |
|---|---|
| DOE | United States Department of Energy |
| EERE | Office of Energy Efficiency and Renewable Energy |
| RPS | Renewable Portfolio Standards |
| R&D | Research and Development |
| D | Rotor Diameter |
| MW | Megawatt |
| W | Watt |
| KM | Kilometer |
| M | Meter |
| U.S. | United States |
| MSE | Mean Squared Error |
| TFBT | TensorFlow Boosted Trees |

| | |
|---|---|
| MAE | Mean Absolute Error |
| RMSE | Root Mean Squared Error |
| $R^2$ | r-squared |
| DFC | Directional Feature Contribution |
| CONUS | Conterminous United States |
| USWTDB | U.S. Wind Turbine Database |
| GAP | Gap Analysis Project |
| ACC | Urban accessibility measured through travel time to nearest urban center (minutes) |
| BUI | Built-up intensity of residential and commercial buildings ($m^2$) |
| CLF | Fractional areal extent of cliff landform (unitless) |
| CON | Landscape metric describing contagion (i.e., the spatial "clumpiness" of unsuitable lands; wind exclusions) |
| CRG | Fraction of pixel containing ridge landform (unitless) |
| DIV | Fractional areal extent of divide landform (unitless) |
| GAP | GAP 1&2 status protected lands (%) |
| HUD | Housing unit density (units/$km^2$) |
| LBL | Wind regions defined by Lawrence Berkeley National Laboratory |
| LC1 | Fractional areal extent of water land cover class (unitless) |
| LC2 | Fractional areal extent of developed land cover class (unitless) |
| LC3 | Fractional areal extent of barren land cover class (unitless) |
| LC4 | Fractional areal extent of forest land cover class (unitless) |
| LC5 | Fractional areal extent of shrubland cover class (unitless) |
| LC7 | Fractional areal extent of herbaceous land cover class (unitless) |
| LC8 | Fractional areal extent of planted/cultivated land cover class (unitless) |
| LC9 | Fractional areal extent of wetlands land cover class (unitless) |
| LFR | Landform regions |
| LSP | Fractional areal extent of lower slope landform (unitless) |
| LU1 | Land use class (level I) |
| MWS | Mean long-term wind speed at 80 m hub height (m/s) |
| POP | Population density (persons/$km^2$) |
| RDG | Fractional areal extent of ridge landform (unitless) |
| RIX | Fractional areal extent exceeding critical slope threshold (unitless) |
| SLF | Fractional areal extent of suitable landforms (i.e., not cliff or valley; unitless) |
| TRC | Fractional areal extent of tree cover |
| USP | Fractional areal extent of upper slope landform (unitless) |
| VLY | Fractional areal extent of valley landform (unitless) |
| WEQ | Power equitability metric describing the evenness of wind energy contributions at 100 m hub height across compass directions |
| WEX | Fractional areal extent of wind exclusions |
| WND | Dimensionless wind energy at 100-m hub height (unitless) |

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
