# Peer review of "Spatially-Explicit Prediction of Capacity Density Advances Geographic Characterization of Wind Power Technical Potential"

_energies, doi:10.3390/en14123609_

Round 1
Reviewer 1 Report
The article describe spatially-explicit prediction of capacity density advances geographic
characterization of wind power technical potential. The topic is important.
General comments:
The lines should be numbered fo reviewers
Please add the list of abbreviatons – it is standard in Energies – e.g. page 14 there are many abbreviations
Detailed comments:
Page 1 - (Beiter et al. 2021) – rather [1] and the rest of references if Energies prefer this type.
Page 11 - at 80m ->at 80 m
Page 11 “An asterisk (*) indicates the varia-ble was selected for use in the final predictive model after screening candidate variables based on correlation coefficient.” – rather below Table 2 not under Table 2
Page 12 “We created a tune grid defining the extent of our hyperparameter space focusing on number of trees (100–1,000), maximum tree depth (3–14) and learning rate (0.001–0.050).” What were other hyperparameters? Who did not you check tree depth = 2? GBT is machine learning technique using small tree depth. Why did you tested 14 tree depth?
Why learning rate checked was so small values?
Page 13 “Variables with correlation > 0.4 were dropped,” What about e.g. minus 0.7 correlation – is it bad input data? Absolute you missed?
MAE, RMSE, R2 – please define
Page 17 “DrivingCcapacit” ????? space needed, Cc???
There are points:
“6. Conclusion and Applications”
“6.1. Applications”
so – where is “Conclusion”? reorganize titles or add 6.1 Conclusion
Reviewer 2 Report
The capacity density of wind energy is a very important factor when calculating wind energy potential, and this paper presents a very reliable statistical analysis result.
However, it is ambiguous whether the 2.74 MW/skm suggested by the authors is simply a result of statistical analysis or a result of machine learning.
If it is a machine learning result, then it is necessary to suggest or translate to a mathematical expression (capacity density = function of variables in Table 2 for instance) rather than Figures 5 and 6.
It is also necessary to discuss whether the results are limited to the United States or can be extended to other countries.
Reviewer 3 Report
I would like to thank the editor for providing the opportunity to review the present manuscript. The manuscript is novel, interesting and well structured, so the reader has no queries. The methods and the results are clearly explained and the literature cited is relevant to the study.
My only concerns about the paper are the first person pronouns in the text, but its up to authors if they want to change it or not.
Moreover, there is a misspell in "4.4. Mechanisms DrivingCcapacity Density Predictions", it is a small mistake but it could be removed since the paper quality is excellent.
In my opinion, the paper should be accepted in present form.
Reviewer 4 Report
Summary:
The authors apply a machine learning model to existing wind farms and geographic characteristics to predict wind capacity density at a 2-km grid spacing across CONUS. Such a map is novel, in that most studies of wind resource do not account for variability of geospatial parameters across large regions. High-resolution datasets like this that predict the spatial variability of the wind resource are valuable resources for energy planning.
Recommendation:
Minor Revision. This article was truly probably the cleanest, most well-written article I have ever reviewed in a first review. It was a pleasure to read! I have only a small number of truly minor comments that ought to be addressed, at which point I think it will be ready for publication. I think this study is novel and a quality contribution to the literature in this field. I commend the authors for the effort they clearly invested to produce an article that is already such high quality at initial submission. I look forward to seeing it in print soon.
Minor Comments:
- Sec. 1.1 and Sec. 6.1 headings: Either create two or more subsections in these sections, or simply remove the 1.1/6.1 headings. (I think simply removing these subsection headings entirely would be best.)
- Ensure that all instances of “i.e.” or “e.g.” have a comma after them (i.e., make it like this).
- Sec. 2.1: After the first instance of “wind power density” in quotes, subsequent uses of the term do not need quotes.
- Sec. 2.2, p. 5: “they are conventionally considered to be homogeneous across space” — What scale(s) do you mean here? Global? Continental? Regional? I suggest changing it to something like “…across large areas, as discussed further below.”
- Sec. 2.3, p. 5: “between -60% and 80%” — If you mean that you varied the estimates between 60% less and 80% more than the reference scenario, then state “+80%” to make that more explicit. As it stands, it sounds like the upper end of the range you tested is merely 80% of the reference value, which I do not think is the intended meaning.
- Sec. 3.1, p. 9: “In total, we excluded from this analysis 282 wind farms representing individual phases of development (28,047 MW).” — Were all phases of multi-phase wind farms excluded? If so, why not retain either the first or last phase? Please explain briefly in the text.
- Sec. 3.6.1, p. 12: Did you try other train/test partitions like 67%/33%, or 50%/50%? How sensitive are your results to the train/test partitioning?
- Fig. 7 and accompanying discussion: This is not a request to change anything, but it is honestly surprising to me to see so many areas of high predicted capacity density in the Southeast, especially in comparison to such low predicted capacity density in the southern Great Lakes region. That is not at all what I would have expected. I appreciate your discussion in Sec. 4.4 that suggests why that is.
- Fig. 8: I suggest that you choose different line/dash patterns for improved colorblind accessibility.
- Sec. 4.4, p. 18: When discussing the ACC metric, please define what counts as a city in your analysis/procedure.
- Sec. 4.4, p. 18: “wind turbines tend to be more closely spaced in forested landscapes than in open (e.g., agricultural) landscapes” — Why is that? This statement is not intuitive to me. I would appreciate you adding at least one or two sentences here in the text explaining why turbines are more tightly packed in forested areas.
- Sec. 6.1, p. 22: Delete the comma after “shape technical potential” and uncapitalize “Federal”.
- Ref. 2: Much of this reference is missing.
